# Dietary Exposure to Glutamates of 2- to 5-Year-Old Toddlers in China Using the Duplicate Diet Method

**DOI:** 10.3390/foods12091898

**Published:** 2023-05-05

**Authors:** Yanjun Zhou, Haixia Sui, Yibaina Wang, Ling Yong, Lei Zhang, Jiang Liang, Jing Zhou, Lili Xu, Yanxu Zhong, Jinyao Chen, Yan Song

**Affiliations:** 1West China School of Public Health and West China Fourth Hospital, Sichuan University, Chengdu 610041, China; zhou__yanjun@163.com; 2Key Laboratory of Food Safety Risk Assessment, National Health Commission of the People’s Republic of China, China National Center for Food Safety Risk Assessment, Beijing 100022, China; 3Shanghai Institute of Quality Inspection and Technical Research, Shanghai 200233, China; 4Heilongjiang Provincial Center for Disease Control and Prevention, Harbin 150030, China; 5Guangxi Center for Disease Prevention and Control, Nanning 530028, China

**Keywords:** glutamates, glutamic acid, dietary exposure, duplicate diet, UHPLC–MS/MS

## Abstract

A duplicate diet collection method was used to estimate dietary exposure to glutamates in children aged 2–5 years in selected provinces of China. Daily duplicate diet samples were collected from 86 healthy toddlers over three consecutive days. Glutamates were analyzed using ultra-high-pressure liquid chromatography–MS/MS (UHPLC–MS/MS). Results showed that the highest glutamates content was found in mixed meals, at 5.12 mg/kg, followed by powdered formula (3.89 mg/kg), and milk and dairy products (2.29 mg/kg). The total mean daily dietary exposure for subjects was 0.20 mg/kg BW, and P95 daily dietary exposure was 0.44 mg/kg BW, both below the acceptable daily intake (ADI) (120 mg/kg BW) recommended by the Joint (FAO/WHO) Expert Committee on Food Additives (JECFA) and the ADI (30 mg/kg BW) set by the European Food Safety Authority (EFSA). Hence it can be considered that glutamates exposure would cause low risk in this group.

## 1. Introduction

Glutamic acid is a non-essential acidic amino acid that occurs naturally in food and presents in the human body while we are still in the womb. Its salt derivatives have a flavor-enhancing effect, first derived from kelp by Japanese scientist Ikeda Kikunae. The mechanism is that glutamate receptors exist on the taste bud cells of the human tongue, which can sense umami taste [1]. Adding glutamates can increase the palatability of food, and, although the results are still inconclusive, some research suggests that glutamates may be addictive, so then consumers may be more inclined to choose them [2,3], and sellers may be more willing to add glutamates in foods to create the umami taste [4]. After oral ingestion, glutamates are mainly absorbed and rapidly metabolized in the small intestine, where they are transported through the sodium transport system and broken down into free amino acids and small molecule peptides, mainly in the intestine, which continue to hydrolyze into free amino acids in the intestinal mucosa cells [5,6,7,8,9,10]. They are also metabolized in the liver, muscle, and brain tissues, decompose into glucose, lactate, glutamine, and other amino acids in the liver [11], and breakdown into alanine and glutamine in muscle tissue [4]. Studies have shown that it is difficult for glutamates to cross the blood–brain barrier [12,13,14,15], and the part that passes through can be metabolized to aspartate in brain tissue [16,17,18,19,20,21]. The metabolites formed are eliminated mainly by deamination, with a small amount excreted by the kidneys [22,23].

Glutamates are approved for use as a food additive in China, the European Union, the USA, Australia, New Zealand, Japan, etc. The Codex Alimentarius Commission (CAC) in CODEX STAN 192-1995 (2019) states that L-glutamic acid and its sodium, potassium, calcium, ammonium, and magnesium salts can be used in appropriate amounts in more than ten food products, including cereals and their products, bakery products, meat and meat products, aquatic products and their products, eggs and egg products, and condiments, as required [24]. The EU Food Additive Regulation (EC) 1333/2008 states that L-glutamic acid and its sodium, potassium, calcium, ammonium, and magnesium salts can use in dozens of food products such as dairy products, oils and fats, edible ice, fruits and vegetables, cereals and starch products, meat products, aquatic products, processed egg products, condiments, beverages, snacks, processed nuts, and dietary supplements at a maximum use level of 10 g/kg, and condiments can be added in moderation [4]. Currently, China only approves the addition of monosodium glutamate (MSG) in glutamate salts as a freshness enhancer in appropriate amounts in foods such as fruit products, vegetable products, soybean products, meat products, aquatic products, egg products, processed nuts, beverages, condiments, and snacks. According to data provided by the China Biotech Fermentation Industry Association, the total global production of monosodium glutamate is currently about 3.5 million tons per year, with Asia leading the world in production and sales, accounting for about 90%, with China accounting for more than 80% of production—2.87 million tons, as of 2019.

Since the introduction of glutamates into the food market, the safety of their use as a freshness enhancer has been evaluated by various national regulatory authorities. The Joint (FAO/WHO) Expert Committee on Food Additives (JECFA) evaluated glutamates in 1970 and 1973, then set the acceptable daily intake (ADI) at 0–120 mg/kg BW (calculated in glutamic acid) [25,26]. However, in 1987 and 2004, it was concluded that the current intake levels indicated by the available evidence did not require further ADI restrictions [27]. Other authorities such as the U.S. Food and Drug Administration (U.S. FDA), the Federation of American Societies for Experimental Biology (FASEB), the Scientific Committee on Food (SCF), and the Food Standards Australia New Zealand (FSANZ) also concluded that there was no need for a limit on glutamates consumption [28,29,30,31,32].

As the demand for glutamates use increases each year, the European Food Safety Authority (EFSA) re-evaluated the safety of glutamates in July 2017. Results showed that glutamates do not induce acute toxicity, subacute toxicity, subchronic toxicity, chronic toxicity, reproductive/developmental toxicity, or other toxic effects in humans. However, this assessment took into account the potential neurotoxicity of glutamates. Based on the available evidence from animal studies, the EFSA Panel finally set the ADI for glutamates at 30 mg/kg BW (calculated in glutamic acid), using the no observed adverse effect level (NOAEL) of 3200 mg/kg BW from a neurodevelopmental toxicity test in rats conducted by Vorhees et al. [33] and an uncertainty factor of 100. The EFSA report also pointed out that the daily intake for infants and children was generally higher than that for the general population and the elderly [4]. Recent studies have shown that the limit of this value is unreasonable [34,35,36].

Animal studies suggested that young organisms were more susceptible to the neurotoxic effects of glutamates. Epidemiological evidence showed that exposure in the younger age groups was different from other groups. Since the use of MSG is probably more common in China, to evaluate the actual intake levels of glutamates in the younger age groups in China, this study first used a duplicate diet method on three consecutive days, which means directly measuring replicate samples of all foods consumed by the participants over a specific period to estimate the exposure to the measured substance accurately [37]. We assessed three regions and six kindergartens in China for dietary glutamates exposure of toddlers in the 2–5 years age group. The daily exposure obtained was then compared with the ADIs set by JECFA and EFSA to assess the estimated daily intake risk of glutamates among participants. In addition, our study roughly estimated the contributions of different food categories (especially powdered formula) to the total dietary glutamates exposure. It is the first study in China and Asia to investigate glutamates exposure levels in toddlers using a duplicate diet collection method, and as far as we know, no similar studies have been conducted in other regions. In addition, our study focused on young children, assessed the glutamates content of formula powder separately, and used the most accurate food consumption survey method to describe the glutamates exposure level in this group more accurately compared with existing reports.

## 2. Materials and Methods

### 2.1. Subjects and Study Design

A three-stage stratified sampling method was used to recruit participants. In the first stage, taking into account geographical differences, economic development, consumption levels, and population size, two cities (Qinzhou and Chengdu) in the southern provinces (Guangxi and Sichuan) and one city (Mudanjiang) in the northern provinces (Heilongjiang) were selected as survey areas. In the second stage, one administrative district was randomly selected in each city, and two kindergartens were randomly selected in each district. In the third stage, 40 healthy participants were selected from each of the southern and northern provinces, i.e., 10 healthy toddlers aged 2–5 years were randomly selected from each of the four kindergartens in the southern province and 20 from each of the two kindergartens in the northern province, making a total of 80 participants. Ultimately, 86 healthy toddlers were included in the survey.

According to Ethical Review Measures for Biomedical Research Involving Human Subjects (2016), issued by the National Health Commission of the People’s Republic of China, dietary studies of food additives conforming to national standards can be ethically exempted. This ethics exemption was approved by the Ethics Committee of the China National Centre for Food Safety Risk Assessment.

### 2.2. Sampling of Duplicate Diets and Preparation

According to the guidance method established by the World Health Organization (WHO) in the corresponding guidelines [38], repeat diet samples were collected from participants for 24 h over three consecutive days. Their parents or guardians provided basic information sheets about the participants and participated in a training session on collection of the samples before the survey. During the survey, parents and guardians were asked to maintain their child’s dietary habits and to replicate as accurately as possible the actual dietary intake at home. Duplicate samples of dietary intake at the kindergarten were collected by trained enumerators. Duplicate food and drinks were collected in containers, with the non-edible parts of the food removed, and subsequently stored in the refrigerator.

Each participant’s basic information, including name, gender, and date of birth, and daily repeat diet samples were divided into mixed meals, milk and dairy products, powdered formula, and drinking water, all collected separately as separate composite samples. Food consumption data for the three days were collected through a food intake diary, which listed all items of solid and liquid food consumed throughout the day. The names and weights of all food items were recorded by the enumerators.

It is worth mentioning that mixed meal samples for each participant needed to be collected daily during the survey. Samples of milk and dairy products and formula powder were collected based on whether or not they were consumed. In addition, each participant’s water intake was recorded and collected daily, and, as the kindergartens provided the participants with identical servings of drinking water, the amount of water collected from each kindergarten was used as the final representative amount.

A total of 89 samples of mixed meals, 101 samples of milk and dairy products, 21 samples of powdered formula, and 6 samples of drinking water were eventually obtained. The daily mixed meal samples were homogenized in the laboratory immediately after the addition of water and stored at −20 °C for analysis.

### 2.3. Analysis of Monosodium L-Glutamate in Food Samples

The method of determination of free monosodium L-glutamate in food by ultra-high performance liquid chromatography–tandem mass spectrometry was validated by the China National Center for Food Quality and Safety Supervision, National Institutes for Food and Drug Control, and the China National Research Institute of Food & Fermentation Industries CO., LTD, to ensure the accuracy and reliability of the results. They are both ISO certified and the first two have qualification certificates issued by the China National Accreditation Service for Conformity Assessment (CNAS). Meanwhile, the units participating in the testing are the inspection institutions at or above the provincial level and have passed the national analytical quality certification. Before data collection, all units passed the unified methodology and experimental technology training. During the sample testing, all units passed the unified blind sample assessment, including the quality control sample assessment and the inter-laboratory comparison review of 2–3% food samples. The method is briefly described as follows.

For foods that do not allow the addition of monosodium glutamate (alcoholic beverages, candy, fruits and their products, milk and milk products, etc.), samples (1.0 g) were loaded into a 10 mL graded tube and then 4 mL of 0.1 M HCl was added to extract L-MSG according to the principle of similar phase solubility. For other foods (cereals, legumes, potatoes, meats, eggs, aquatic foods, and vegetables), an additional 1.0 g of NaCl was added for extraction, because it does not interfere with the detection of monosodium glutamate, and, compared with other reagents, NaCl significantly reduces the degree of emulsification after high-speed centrifugation. The vortex was 2 min and centrifuged for 5 min (10,000 rpm). Samples in which glutamate was not detected were used as blank samples.

The liquid chromatography system (Shimadzu Corporation, Kyoto, Japan) comprised a system controller (CBM-20A), an automatic sampler (SIL-30AC), a column temperature chamber (CTO30A), and a conveyor pump (LC-30AD). The triple-quadrupole tandem mass spectrometer was from Shimadzu (Japan). The temperature of the column oven was 40 °C. Water (phase A) and ACN (phase B) containing 0.2% formic acid (*v*/*v*) served as mobile phases. The gradient elution program involved two steps: 0–3 min: 50% phase B; 3–6 min: phase B content increased linearly to 100% and maintained for a further 2 min. The flow rate was 0.20 mL/min and the injection volume was 1 μL.

The ionization mode was positive-ion mode electron spray ionization (ESI). The ion spray voltage was +4.5 kV. Nitrogen at flow rates of 3 L/min and 10 L/min was utilized as the atomization and drying gas. The heating gas (air) velocity was 10 L/min. The collision gas was argon. The desolvation line (DL) and heating module temperatures were 300 °C and 400 °C, respectively. Dwell and delay times were 8 ms and 3 ms, respectively. The retention time of L-MSG was 2.714 min. Quantitative and qualitative ion pairs of L-MSG in MRM scanning mode were 147.90 > 84.05 and 147.90 > 56.05, with collision energies of 16 eV and 23 eV, respectively.

L-MSG (0.01 g) was weighed and dissolved in 50% aqueous ACN (*v*/*v*) in a 10 mL volumetric flask to produce a 1000 mg/L standard stock solution. Then, 100 mg/L and 10 mg/L standard working solutions were prepared by diluting aliquots of the stock solution with 50% aqueous ACN (*v*/*v*). All the solutions were stored at –20 °C. The standard curves were prepared using 0.01 g L-MSG standard solutions at 0.001 mg/L, 0.01 mg/L, 0.02 mg/L, 0.05 mg/L, 0.1 mg/L, 0.2 mg/L, 0.5 mg/L, and 1.0 mg/L. The peak areas were the ordinates, with the concentrations of l-MSG as the abscissae.

The absolute difference between two independent determinations obtained under reproducible conditions should not exceed 10% of the arithmetic mean, with a limit of detections (LODs) of 1–3 μg/kg and a limit of quantifications (LOQs) of 5–10 μg/kg. The non-detects content was assigned a value of 1/2 LOD in this assessment by the principles in the second meeting Credible Assessment of Low-Level Contaminants in Foods of the WHO GEMS/FOOD.

According to the document “Analytical Detection Limit Guidance”, issued by the Wisconsin Department of Natural Resources Laboratory Certification Program, the method detection limits (MDLs) and practical quantification limits (PQLs) of this method were obtained by spiking matrices with standards in our study. The lowest spiking levels in the matrices giving RSDs of 30% and 15% were taken as the MDLs and PQLs. In three of the matrix spiking level, 10 μg/kg, 100 μg/kg, and 1000 μg/kg, the recoveries and RSD% were 77.0–102.0% and 2.3–6.0, 80.1–122.1% and 2.2–9.7, and 95.6–119.0% and 2.2–14.3, respectively.

### 2.4. Exposure Assessment

Assuming 100% hydrolysis of monosodium glutamate in the replicate food samples, the estimated dietary intake was calculated using the simple distribution method based on data on the daily consumption of food samples from the target population and the concentration of MSG in the food groups surveyed, using the following equation:$${Exp}_{i}=\sum_{i=1}^{n}\frac{F_{ik}\times C_{ik}}{{BW}_{i}}\times 0.8698÷1000$$
where *Exp_i_*—consumer i’s exposure to glutamates from all duplicate diet samples, mg/kg BW;

*F_ik_*—the level of individual consumption of consumer *i* from food *k*, g/day;

*C_ik_*—the concentration of monosodium glutamate in the replicate food sample *k* for consumer *i*, mg/kg;

*BW_i_*—the individual weight of consumer *I*, kg;

0.8698—the conversion factor for the conversion of monosodium glutamate to glutamates meter;

1000—the unit conversion; and

*n*—the amount of food consumed by consumer *i*.

Exposures were specified as the mean, 50th percentile, 95th percentile, and range of the intake distribution for all consumers for males and females, respectively.

### 2.5. Risk Characterization

The potential health risk of glutamates was assessed by comparing the estimated daily exposure levels to glutamates per unit body weight with the ADI recommended by JECFA (120 mg/kg BW) in 1973 and ADI (30 mg/kg BW) recommended by EFSA in 2017.

### 2.6. Statistical Analysis

Data were analyzed using the statistical package SPSS 26 (SPSS, Chicago, IL, USA). Descriptive statistical analyses were performed for incidence data, food consumption data, and exposure data. Differences between subgroups were tested using the independent samples t-test and ANOVA. *p* < 0.05 was considered statistically significant.

## 3. Results

### 3.1. Participants Characteristics

The study population consisted of 46 boys and 40 girls, aged between 26 and 68 months (mean: 39.5 months for males and 39.7 months for females), weighing between 11.0 and 20.0 kg (mean: 16.0 kg for males and 15.4 kg for females), and with a height between 87.0 and 115.1 cm (mean: 99.0 cm for males and 98.0 cm for females). There were 49 participants in the 2–3 years age group, 21 in the 3–4 years age group, and 16 in the 4–5 years age group.

### 3.2. Glutamates Concentrations in Duplicate Food Samples

In this study, glutamates levels were analyzed in mixed meal samples including grains, meats, aquatic, vegetables, fruits, eggs, beverages, and other foods with their products, milk and dairy products samples, and powdered formula. As shown in Table 1, the mean glutamates level was 5.12 mg/kg for mixed meals samples (*n* = 87), 2.29 mg/kg for milk and dairy products (*n* = 105), and 3.89 mg/kg for powered formula samples (*n* = 49). Glutamates were not detected in all drinking water samples (*n* = 6).

### 3.3. Estimated Daily Exposure to Glutamates from Repeated Food Samples

The estimated daily exposure to glutamates in toddlers was calculated using the glutamates content and consumption data of duplicate food samples obtained from the survey, as well as individual weight.

The mean value of the mixed meals consumed by the study population during the survey period was 859.54 g/d, with male and female participants consuming an average of 754.94–745.62 g of mixed meals per day (range: 428.30–1427.30 g), with no significant difference between gender groups (*p* > 0.05). Among the various food groups, grains were the most consumed food categories, accounting for 50.01%, followed by beverages and others, with 21.56% and 20.63%, respectively, followed by vegetables and fruits, with 16.71% and 15.08%, respectively, and the meats, eggs, and aquatic with 6.95%, 5.16%, and 2.40%, respectively. This distribution was found in each age group, with no significant differences between groups (*p* > 0.05), as detailed in Figure 1.

The mean value of consumption of milk and dairy products for the total participants was 162.46 g/d, with male and female participants consuming an average of 128.21–124.36 g of dairy products per day (range: 40.10–418.50 g). The mean consumption of powdered formula was 33.47 g/d, with male and female participants consuming an average of 34.96–33.53 g per day (range: 10.50–88.10 g). The highest consumption of milk and dairy products was in the 3–4 years age group, with a mean value of 204.31 g/d, and the lowest consumption was in the 2–3 years age group, with a mean value of 139.15 g/d. The highest consumption of powdered formula was in the 3–4 years age group, with a mean value of 36.80 g/d, and the lowest consumption was in the 4–5 years age group, with a mean value of 30.20 g/d, as shown in Table 2 and Table 3.

As shown in Table 4 and Figure 2, the estimated mean daily exposure for glutamates ranged from 0.20 to 0.37 mg/kg BW, P95 daily exposure ranged from 0.44 to 0.63 mg/kg BW, and the polar range was 0.03–0.68 mg/kg BW.

The mean daily exposure for toddlers in the 2–3 years age group was 0.20 mg/kg BW, P95 was 0.44 mg/kg BW, and the extreme range was 0.03–0.51 mg/kg BW. The mean daily exposure for the 3–4 years age group was 0.27 mg/kg BW, P95 was 0.50 mg/kg BW, and the extreme range was 0.10–0.57 mg/kg BW. The mean daily exposure for the 4–5 years age group was 0.37 mg/kg BW, P95 was 0.65 mg/kg BW, and the extreme range was 0.13–0.68 mg/kg BW.

No statistically significant differences were found between age and gender groups when ANOVA tests were conducted based on previous statistical analyses (*p* > 0.05).

The range of glutamates exposure showed that the exposure level in toddlers increased with age, but there was no significant difference among age groups (*p* all > 0.05). All were seen to be well below the JECFA ADI (120 mg/kg BW), as well as the ADI (30 mg/kg BW) set by EFSA. In addition, 97.37% of the mean dietary exposure to glutamates was from mixed meals, 10.26% from milk and dairy products, and 2.81% from the powdered formula, with a mean exposure of 0.01 mg/kg BW. The corresponding mean and 95th percentile exposure values for each food group are detailed in Figure 3.

## 4. Discussion

Glutamates are widely used as a food additive in a variety of foods for their freshness-enhancing effects. Following the ADI limit by EFSA in 2017, concerns have been raised about the safe range of its daily use, as well as the developmental neurotoxicity of infants and children. To this end, we performed a risk assessment and finally determined the same ADI as determined by JECFA, i.e., 120 mg/kg BW (calculated in glutamic acid), according to the actual situation of Chinese residents (data not published yet). Results showed that the daily glutamates exposure of Chinese 2–5 year olds was much lower than the ADI recommended by JECFA. Therefore, although the daily glutamates exposure data obtained from this survey were compared with the ADI set by JECFA, to make full use of the existing data, the following section compares the information reported by EFSA and other related research to evaluate the exposure levels of glutamates in the Chinese population aged 2–5 years.

We conducted the duplicate diet collection method in this study, which reflected the food consumption situation of toddlers more accurately. In addition, this study collected the consumption data of milk and dairy products, and formula powder separately to assess glutamates exposure levels for the first time. However, due to the difficulty of implementing this method, there existed an incomplete collection of food samples during the investigation. Meanwhile, the sample size was not large enough, which might result in poor representation or incomplete utilization of consumption data. At the same time, the content detection method we used cannot distinguish naturally occurring and artificially added glutamates, considering that the natural glutamates content of meats, milk and dairy products, and the powdered formula is usually higher, which may have led to an overestimation of glutamates exposure levels derived from these foods.

In our study, the consumption of grains was the highest in the mixed meals of Chinese toddlers aged 2–5 years, accounting for 50.01%, followed by beverages and other foods, accounting for 21.56% and 20.63%, respectively, then vegetables and fruits, accounting for 16.71% and 15.08%, respectively, with the average consumption of these five food groups ranging from 129.63 to 429.85 g/d. The consumption level of milk and dairy products was located between other categories and vegetables, with a mean value of 162.46 g/d; the mean value of powdered formula consumption was 30.20 g/d, ranking the second lowest. Compared with other Asian countries, investigations showed that the consumption of grains, meat, vegetables, and fruits of children aged 4–6 in South Korea is 332.8 g/d, 42.1 g/d, 93.1 g/d and 138.7 g/d [39], respectively, while the consumption of aquatic products and dairy products of children aged 3–6 in Japan is 21.14 g/d and 128 g/d, respectively [40]. The consumption proves similar to that of the children in China above, so it can be considered that the data obtained in this survey can represent the consumption situation in Asia.

The results showed that the mixed meals contained the most glutamates, followed by powdered formula, and milk and dairy products, with mean values of 5.12 mg/kg, 3.89 mg/kg, and 2.29 mg/kg respectively. Because this study did not examine the individual glutamates content of the classified foods in the mixed meals, only dairy products and powdered formula were compared with the values provided by EFSA. The powdered formula tested in this study was compared with 13.1.1 (infant formulae as defined by Commission Directive 2006/141/EC) and 13.1.2 (follow-on formulae as defined by Directive 2006/141/EC) of the Food Classification System (FCS) provided in the report. Milk and dairy products correspond to 01.4 (flavored fermented milk products including heat-treated products) in the FCS. As can be found in the Annex of the report, the final EFSA average values for the corresponding levels are 2952 mg/kg, 2335 mg/kg, and 72 mg/kg, respectively, based on the information provided by the Member States and the exposure scenarios at the time of consumption [4].

In its report, EFSA calculated a total of three different scenarios. Considering the daily exposure of different populations, we chose a third scenario closer to the study context for comparison. The reported mean estimated daily exposure values for children aged 2–5 years in the Member States ranged from 22 to 158 mg/kg BW, the 95th percentile exposure values ranged from 38 to 429 mg/kg BW [4], and the exposure ranges obtained in this study were 0.20–0.37 mg/kg BW and 0.44–0.65 mg/kg BW, respectively. The contribution rates of mixed meals, milk and dairy products, and the powered formula were 97.37%, 10.26%, and 2.81%, respectively.

In summary, we can see that the exposure levels obtained in this study are lower than the values obtained in the EFSA report, and there exists a large gap. We believe that this may be due to several factors. The first point is the difference between glutamates content in foods. The countries for which data are available in the EFSA report include Germany (*n* = 31,758), Spain (*n* = 304), Hungary (*n* = 302), the United Kingdom (*n* = 296), Luxembourg (*n* = 9), and the Czech Republic (*n* = 2), which, except for the United Kingdom, are all European Union member states. In addition, the UK still continue to use the EU regulations of glutamates. Therefore, the food levels of glutamates in these countries are most likely higher than those in China, as they allow the addition of glutamate, monosodium glutamate, monopotassium glutamate, calcium di-glutamate, ammonium glutamate, and magnesium di-glutamate, whereas China only allows the addition of glutamate and monosodium glutamate. At the same time, EFSA did not describe the testing methods used in the reporting countries and the types of food tested. These two factors are more likely to account for the difference in exposure levels, given that these countries primarily use monosodium glutamate, similar to China. Since the existing studies on glutamates usage are all from EFSA reports, other countries such as the United States have not reported their usage level. Therefore, further research is needed to determine the source of the difference in content.

The second point is that the countries that provided data did not include countries in the Asian region, where there are differences in consumption habits and dietary consumption. Compared with European countries, Asian countries consume a higher proportion of cereals and plant-based foods and a lower proportion of animal foods, which usually contain more glutamates than the first two type of foods. In addition, although Asian countries have a higher proportion of certain foods in their consumption habits, the total dietary consumption level of European countries is still higher than that of Asian countries. In Germany, for example, where the highest percentage of data was provided (97.21%), the average consumption of vegetables for children aged 3–6 years was 340–360 g/d and the average consumption of fruit was 200–220 g/d, according to the National Nutrition Survey (KiGGS) [41], and children aged 3–10 years consumed 626.1–722.1 g/d of beverages [42]; therefore, compared with the consumption of Chinese children aged 3–5 years obtained in this study (143.64 g/d, 129.63 g/d, 185.31 g/d, respectively), their consumption appears to be higher, which may also lead to a higher estimated daily glutamates exposure in the EFSA final calculation.

Therefore, considering that the duplicate diet collection method is one of the most accurate dietary survey methods available, it is worth exploring whether the geographical difference caused the large discrepancy between the glutamate levels obtained in this study and the estimated daily exposure calculated from the EFSA 2017 report. In conclusion, the results need to be interpreted cautiously considering that the limited sample size of this study may not be fully representative of the actual consumption level in China.

## 5. Conclusions

In summary, this study assessed the level of glutamates exposure in the lower age group in China, and it is the first study to use the duplicate diet method to assess the glutamates exposure level of Chinese toddlers aged 2–5 years. Three categories of foods were analyzed, including mixed meals, milk and dairy products, and powdered formula. The mean daily estimated exposure of this population was 0.20–0.37 mg/kg BW, and P95 of 0.44–0.63 mg/kg BW, which was much lower than the ADI of 120 mg/kg BW set by JECFA and lower than the ADI of 30 mg/kg BW recommended by EFSA (both calculated in glutamic acid). It can be assumed that the glutamates intake of Chinese toddlers through daily diet would not induce a health risk. However, it is worth mentioning that the sample size of this survey is limited, and the glutamates exposure level is quite different from that of European countries. More studies are needed to explore the representativeness of the results of this survey and the reasons for the differences in exposure levels.

## Figures and Tables

**Figure 1 foods-12-01898-f001:**
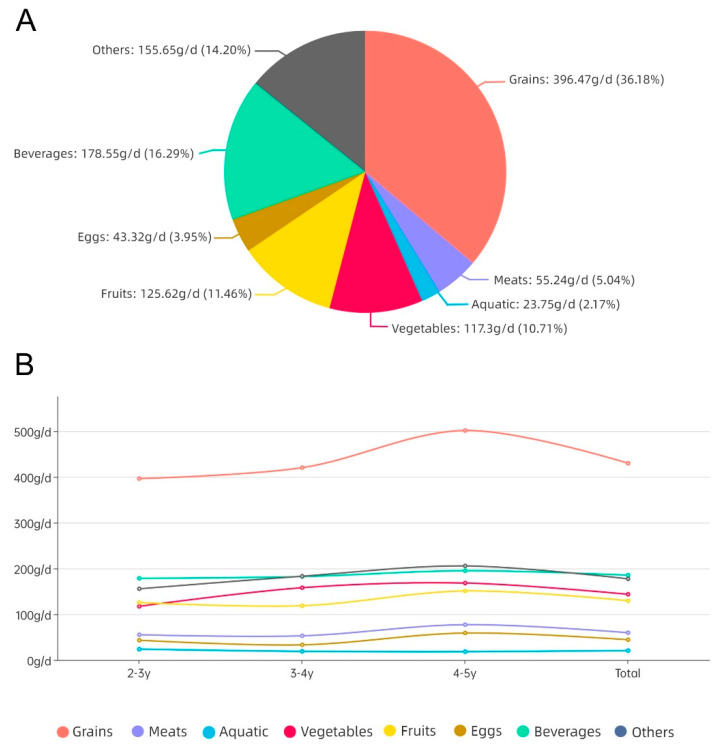
Proportion of total food consumption (**A**), and food consumption trends in different age groups (**B**) among toddlers aged 2–5 years old.

**Figure 2 foods-12-01898-f002:**
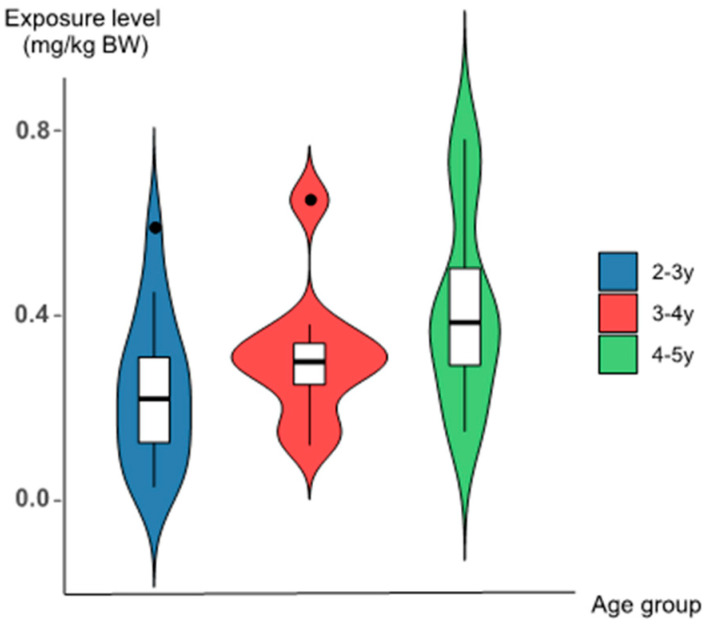
Glutamates exposure level among different age groups.

**Figure 3 foods-12-01898-f003:**
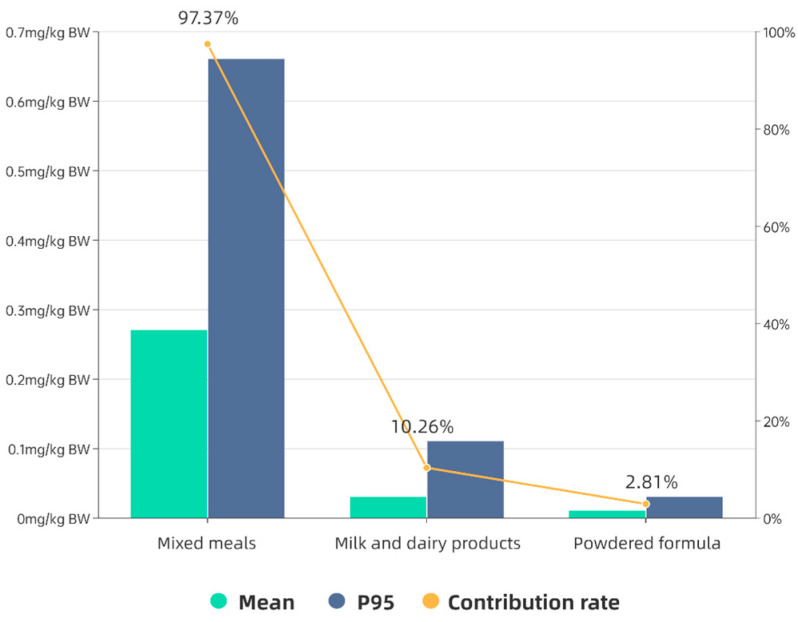
Glutamates exposure level and contribution rate of different food categories.

**Table 1 foods-12-01898-t001:** Glutamates levels in duplicate food samples (mg/kg).

Category	Mean ± SD	P50	P90	P95	P97.5	P99 *
Mixed meals	5.12 ± 4.94	4.03	10.59	15.40	17.98	24.68
Milk and dairy products	2.29 ± 2.17	1.45	5.34	7.18	9.02	10.57
Powdered formula	3.89 ± 3.12	3.24	7.01	10.57	13.89	14.90

* The figure within P indicates how a percentage of the data are lower than this value. For example, P50 means that 50% of the data are lower than this value and 50% are higher; P90 means that 90% are lower than this value and 10% are higher; P95 means that 95% are lower than this value and 5% are higher.

**Table 2 foods-12-01898-t002:** Consumption of milk and dairy products in the 2–5 year olds (g/d).

Age	Mean ± SD	P50	P90	P95	P97.5	P99 *
2–3	139.15 ± 10.35	100.00	274.90	383.92	478.12	619.38
3–4	204.31 ± 18.00	156.70	365.32	419.48	473.63	515.74
4–5	179.00 ± 19.78	139.70	363.60	385.32	405.24	414.87
Total	162.46 ± 117.40	125.00	344.00	404.33	454.38	592.24

* The figure within P indicates how a percentage of the data are lower than this value. For example, P50 means that 50% of the data are lower than this value and 50% are higher; P90 means that 90% are lower than this value and 10% are higher; P95 means that 95% are lower than this value and 5% are higher.

**Table 3 foods-12-01898-t003:** Consumption of powdered formula in the 2–5 year olds (g/d).

Age	Mean ± SD	P50	P90	P95	P97.5	P99 *
2–3	32.95 ± 1.46	32.70	51.70	58.60	62.70	65.69
3–4	36.80 ± 2.99	33.00	60.64	61.55	65.19	70.24
4–5	30.20 ± 1.96	25.60	37.05	41.88	46.94	49.98
Total	33.47 ± 14.00	32.50	56.10	61.10	62.98	82.59

* The figure within P indicates how a percentage of the data are lower than this value. For example, P50 means that 50% of the data are lower than this value and 50% are higher; P90 means that 90% are lower than this value and 10% are higher; P95 means that 95% are lower than this value and 5% are higher.

**Table 4 foods-12-01898-t004:** Daily dietary exposure to glutamates in participants aged 2–5 years (mg/kg BW).

Age	N	Mean ± SD	P50 *	P95 *	Range
2–3	15	0.20 ± 0.04	0.19	0.44	0.03–0.51
3–4	9	0.27 ± 0.04	0.26	0.50	0.10–0.57
4–5	8	0.37 ± 0.07	0.35	0.65	0.13–0.68
Total	32	0.26 ± 0.03	0.26	0.63	0.03–0.68
%ADI (JECFA)	- *****	0.22	0.22	0.53	0.03–0.57
%ADI (EFSA)	-	0.87	0.87	2.10	0.10–2.27

* P50 means that 50% of the data are lower than this value and 50% are higher, P95 means that 95% are lower than this value and 5% are higher. “-“ means no value.

## Data Availability

Data is contained within the article.

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
