# Peer review of "Dietary Exposure to Glutamates of 2- to 5-Year-Old Toddlers in China Using the Duplicate Diet Method"

_foods, 2023, doi:10.3390/foods12091898_

Round 1

Reviewer 1 Report (Previous Reviewer 1)

The authors have addressed my concerns. The reason for addition of NaCl should be included in the experimental section (line 167). I believe the authors mean the significant degree of emulsification after high-speed centrifugation is greatly decreased (not improved) by the use of NaCl.

Author Response

Response to Reviewer 1 Comment

Point 1: The authors have addressed my concerns. The reason for addition of NaCl should be included in the experimental section (line 167). I believe the authors mean the significant degree of emulsification after high-speed centrifugation is greatly decreased (not improved) by the use of NaCl.

Response 1: Thanks for the comment, we have added “because it does not interfere with the detection of monosodium glutamate, and compared to other reagents, NaCl significantly reduces the degree of emulsification after high-speed centrifugation” on page 4, lines 169-171.

Reviewer 2 Report (Previous Reviewer 2)

I believe that the article is an interesting one. Some remarks:

- in the introduction should be mentioned that after line 32 the fact that monosodium glutamate can be addictive, that umami taste is present from the womb mother, that companies are using monosodium glutamate in order to create umami taste;

- to the discussion part I am not agree with the affirmation that becouse EU countries are using different type of glutamates than the level is much higher< the most used by all countries is monosodium glutamate (lines 363-383). Also United Kingdom is not a EU country. More, some informations related to United State of America should be provided. I think that the highest problem related to glutamate exposure is there. I think as authors suppose that consumption habits  is the problem to glutamates exposure. This fact should be presented in the discussion part in an extensive way.

- the conclusion are to short. The authors must underline if the objectives were achieved, to offer us some recommandations, etc.

Author Response

Thanks for the comments, please see the attachment.

Reviewer 3 Report (Previous Reviewer 3)

This manuscript is a re-submission of the authors’ previous manuscript. My point for the previous manuscript was only one: external quality control of glutamate analysis is necessary.

Since one of the conspicuous points of this work is the far lower glutamate intake of the subject population than that in the previous studies, I have been continuously pointing out that the authors must guarantee the accuracy of glutamate analysis. I can find the value of this work only when the lower intake the authors found is free from analytical artifact.

However, the authors did not cope with this. Accreditation of lab, which the authors added in the manuscript in response to my comment on the original submission, does never guarantee the accuracy of glutamate analysis. This was clearly commented in the 2nd revision of the original submission. The authors could have done recovery test of dietary glutamate analysis for the resubmission, but they didn’t.

My evaluation of this manuscript is still the same. Presentation of the results of external quality assurance of glutamate is needed.

There is nothing to comment to other part of the manuscript.

Author Response

Thanks for the comment, please see the attachment.

This manuscript is a resubmission of an earlier submission. The following is a list of the peer review reports and author responses from that submission.

Round 1

Reviewer 1 Report

Based on a Google scholar search for glutamate determination in toddlers, this type of study has not been previously reported. The quality of the work and statistical analysis is good. It would be helpful to remind the reader, for example, what P95 means in the text (95% of the data is lower than this value and 5% is higher) because so many P values are used in the tables. The extraction efficiency using NaCl (typo on line 154) of the samples is not reported. Were some samples spiked with glutamate to understand the recovery? Which samples required the addition of 1 g NaCl and why (lines (152-158). The LC-MS method is sound using anion exchange and formic acid in the mobile phase. Definitions for how LOD and LOQ were determined should be given. 

Reviewer 2 Report

The article is very interesting by it subject. The authors clearly underline the novelty of their study. Some minor suggestions:

- the authors did not mention nothing about umami taste.  It creates addiction and the humans begin to eat more and more foods with this type of taste. Please insert in the introduction part the connection between glutamates and umami taste and it impact on food production and consumption.  Also the references part must be improved with more from the last five years (the majority are very old).  Please insert under the table what P50, P90, P95 mean.  Also please insert standard deviation in tables. An ANOVA to see the significant differences between age groups and gender would be interesting with a more scientific impact. 

Reviewer 3 Report

The work presented in this manuscript is an estimation of daily glutamate intake in toddlers in China based on a duplicated diet study. The glutamate intake was estimated to be much lower than the ADI of EFSA and JECFA. This kind of information, though descriptive, is valuable for the assessment of food safety.

There are a couple of concerns in this study.

The small number of subjects of this study (n=86) is one of the concerns. The authors point out that the present study subjects were sampled by three-stag(Line 102) but the selected number is too small to be representative of China if one consider the diversity in food habit of this country.

The concern in this study result is much lower glutamate intake than the levels found in European countries. In this respect, the authors should present reliability of their glutamate analysis so that the difference between this study and European data is not due to analytical deficiency. Analytical quality control (QC), particularly external QC, of this work should be presented. The manuscript only describes about internal QC (blank, calibration, LOD, or duplicate analysis) but not at all about external QC.

Minor points:

It has to be also pointed out that the manuscript is not very well written: the manuscript must be edited by a native speaker before submission. Equations in page 4 are so fundamental elementary mathematics and there is no need to describe these equations. Delete them.

Round 2

Reviewer 3 Report

External QC is not yet properly described. The authors seems not to be familiar with quality control: detection limit is not an essential part of external QC.

In contrast to the authors' reply, table 1 does not present recovery of added standard. The authors cannot gurantee the accuracy of their measurement in the revised version, and this makes me consider that the low glutamate intake of this population is not free from analytical suspicion.

Author Response

Thank you for your suggestion. We apologize for misunderstanding the question. We mentioned the relevant information on page 4, lines 174-177 of the article but did not explain it in detail.

Three institutions with national analytical quality certification verified this glutamates method. The units participating in the testing are the inspection institutions at or above the provincial level and have passed the national analytical quality certification. Before data collection, all units passed the unified methodology and experimental technology training. During the sample testing, all units passed the unified blind sample assessment, including the quality control sample assessment and the inter-laboratory comparison review of 2%-3% food samples. We have added this description to page 4, line 177.

In addition, we conducted a study of glutamates in adults using this method to calculate daily exposures, which were similar to the results of large population surveys in other Asian regions (Taiwan, Japan, South Korea, Vietnam, etc.). At the same time, we also proved through literature research that the consumption of various types of food in China is similar to those areas. Therefore, we believe that the glutamates content obtained by this detection method is sufficiently credible.